# Stem Cell Origin of Cancer: Clinical Implications beyond Immunotherapy for Drug versus Therapy Development in Cancer Care

**DOI:** 10.3390/cancers16061151

**Published:** 2024-03-14

**Authors:** Shi-Ming Tu, Anup K. Trikannad, Sruthi Vellanki, Munawwar Hussain, Nazish Malik, Sunny R. Singh, Anusha Jillella, Sri Obulareddy, Sindhu Malapati, Sajjad A. Bhatti, Konstantinos Arnaoutakis, Omar T. Atiq

**Affiliations:** Division of Hematology and Oncology, University of Arkansas for Medical Sciences, Little Rock, AR 72205, USAmhussain@uams.edu (M.H.); nmalik@uams.edu (N.M.); srsingh@uams.edu (S.R.S.); ajillella@uams.edu (A.J.); sobulareddy@uams.edu (S.O.); smalapati@uams.edu (S.M.); sabhatti@uams.edu (S.A.B.); karnaoutakis@uams.edu (K.A.); otatiq@uams.edu (O.T.A.)

**Keywords:** immunotherapy, cancer stem cell, immune privilege, immune modulation, drug development, multimodal therapy

## Abstract

**Simple Summary:**

The premise that certain checkpoint inhibitors are effective anti-cancer treatments beyond their immune modulatory capabilities influences current cancer research and clinical practice as well as future drug versus therapy development. We propose ways and means to enhance immunotherapy in cancer care by combining anti-PD1/L1 with various therapeutic modalities according to an appropriate scientific theory, e.g., stem cell origin of cancer, and based on available clinical evidence, e.g., randomized clinical trials.

**Abstract:**

Although immunotherapy has revolutionized cancer care, there is still an urgent need to enhance its efficacy and ensure its safety. A correct cancer theory and proper scientific method empower pertinent cancer research and enable effective and efficient drug versus therapy development for patient care. In this perspective, we revisit the concept of immune privilege in a cancer cell versus normal cell, as well as in a cancer stem cell versus normal stem cell. We re-examine whether effective immunotherapies are efficacious due to their anti-cancer and/or immune modulatory mechanisms. We reassess why checkpoint inhibitors (CPIs) are not equal. We reconsider whether one can attribute the utility of immunotherapy to specific cancer subtypes and its futility to certain tumor/immune compartments, components, and microenvironments. We propose ways and means to advance immunotherapy beyond CPIs by combining anti-PD1/L1 with various other treatment modalities according to an appropriate scientific theory, e.g., stem cell origin of cancer, and based on available clinical evidence, e.g., randomized clinical trials. We predict that a stem cell theory of cancer will facilitate the design of better and safer immunotherapy with improved selection of its use for the right patient with the right cancer type at the right time to optimize clinical benefits and minimize potential toxic effects and complications.


*Science is the belief in the ignorance of experts.*
Richard Feynman.

## 1. Introduction

Although we are more enlightened and less ignorant about the immune system than ever before, the fact is that our understanding and insight are still relatively elementary than advanced. Although we have better cancer immunotherapy at our disposal, the truth is that many of our modern cancer treatments are still rather rudimentary than avant-garde.

To understand cancer immunity, we need to elucidate the responsibilities and liabilities of the immune cells as well as investigate the vulnerabilities and vagaries of the cancer cells. 

To harness the power and ensure the safety of immunotherapy, we need to be selective in its use for the right patient with the right cancer type at the right time and sagacious in its combination with other treatment modalities to optimize clinical benefits and minimize potential toxicities and complications.

A stem cell theory of cancer invites us to revisit the concept of immune privilege in a cancer cell versus normal cell, as well as in a cancer stem cell (CSC) versus normal stem cell (SC), both of which are immune privileged for a singular reason and perhaps for an ulterior purpose (as in the former entity). 

In this perspective, we re-examine whether effective immunotherapies are efficacious due to their anti-cancer and/or immune modulatory mechanisms. We wonder why checkpoint inhibitors (CPIs) are not equal. We ponder about ways and methods to enhance immunotherapy to improve patient care: treating various cancer subtypes vs. certain tumor/immune compartments, components, and microenvironments by combining anti-PD1/L1 with various other treatment modalities based on available clinical evidence, e.g., randomized clinical trials.

## 2. Brief History

According to writings in the Ebers Papyrus (c1550 BC), the great Egyptian physician Imhotep (c2600 BC) recommended “an incision which would lead to an infection of the tumor and its regression” for the treatment of tumors (swellings) [1].

Perhaps Coley took Imhotep’s words to heart after observing spontaneous remission in a patient with sarcoma. The patient had an egg-size mass in his cheek. Although he had had two prior surgeries to remove this tumor, it still recurred and progressed. After another surgery that could only partially remove the grape-like tumor, he developed recurrent infections (erysipelas caused by the bacteria *Streptoccocus pyogenes*) in a wound that would not heal. Surprisingly, after each bout of high fever, the ulcerated wound improved. Gradually, the tumor shrank. Eventually, it vanished. Seven years later, the patient still had a large scar but no trace of any cancer on his face.

Coley is credited with the first scientific immunotherapy even though his next 10 patients treated with erysipelas (1893) did not benefit from the treatment, and his subsequent treatments using “Coley’s toxins or vaccine” under accepted scientific methods did not indicate clinical efficacy [2,3]. 

In the mid-1980s, another milestone in the annals of immunotherapy happened when intravesical BCG was found to be effective for the treatment of superficial bladder cancer [4] and replaced cystectomy as the treatment of choice for carcinoma in situ of the bladder. 

In 2010, the first ever FDA-approved vaccine for cancer care, sipuleucel-T, was demonstrated to provide a modest statistical benefit, i.e., overall survival (OS) improvement of about 4 months for patients with castration-resistant prostate cancer (CRPCa) [5,6].

In 2018, two immunologists, Hojo and Allison, received the Nobel Prize in Physiology or Medicine for their discovery of cancer therapy by inhibition of negative regulation of PD1 and CTLA-4, respectively [7,8]. 

Fortuitously or not, Hojo and Allison have joined or perhaps started a highly anticipated and much touted revolution in cancer care involving immunotherapy in the modern era.

## 3. Immune Privilege

When we talk about immunotherapy, it is essential for us to think about the origin and nature of immune privilege in both cancer and normal cells [9]. According to a stem cell theory of cancer, similarities in intrinsic immune privilege between cancer and normal stem cells may predicate the success or implicate any failure of immunotherapy in cancer care.

There is a fundamental reason and necessity for our normal stem cells to be immune privileged, protected, and preserved. We cannot afford to destroy or damage our normal stem cells, which enable us to proliferate, propagate, and recuperate. Otherwise, we suffer from pro-aging ailments and autoimmune maladies.

If cancer has a stem cell origin and is a stem cell disease, then we face a formidable challenge to overcome this inherent immune privilege with immunotherapy. In other words, if CSCs are derived from normal SCs, they may be endowed with the same immune capabilities and equipped with the same immune safeguards. If CSC mimics and mirrors normal SC, then separating the two is a daunting but prerequisite task. Otherwise, clinical benefits with immunotherapy for some patients may not be commensurate with potential risks, and we may cause more harm than good in the care of certain patients, especially for those who are less or minimally threatened by their cancers. 

We propose that the proper answers to the following pertinent questions may clarify whether we have the right or wrong scientific theory about certain basic mechanisms of immunotherapy in cancer care. 

### 3.1. Does Anti-PD1/L1 Provide Immediate Clinical Efficacy (e.g., within 7 Days)?

One expects that immunotherapy would elicit a delayed onset of action, since immune activation takes time.

Hence, the median time of onset for pembrolizumab is 1.2 months (range: 0.5 to 3.5 months). However, if therapeutic effects do in fact occur early and almost “immediately”, such as within 1 week of treatment [10,11], do they indicate or implicate direct anti-cancer activity rather than immune activation/modulation? 

One may attribute such “exceptional” or “anecdotal” results to lucky events or placebo effects. But one could also ascribe rarity of the event to our inability to detect or report such an event. When we have better means to assess efficacy sooner (e.g., palpable/symptomatic or biomarker improvement [12]) rather than later (e.g., measurable response on imaging that is not normally performed before 4–6 weeks after start of treatment), then it is conceivable that “immunotherapy” would have acted before immune activation, suggesting that some other mechanisms of action, such as direct anti-cancer activity, might have played a role in the therapeutic process instead.

### 3.2. Does a Brief or Bolus Anti-PD1/L1 Treatment Provide Durable If Not Permanent Clinical Benefits?

One would expect that a short duration of immunotherapy could and should provide lasting therapeutic effects, because the immune system has a memory, like a vaccine, the ultimate immunotherapy, does.

Certainly, this expectation may be true for “real” immunotherapies like high-dose interleukin-2 (IL-2) and an anti-CTLA4, ipilimumab, in which some patients who attain a complete response (CR) have managed to maintain their remissions for years, if not decades, after a brief course or short duration of treatment.

However, it may not be true for “pseudo”-immunotherapies, like the anti-PD1, nivolumab, in which continuous treatment seems to provide better clinical outcome compared with fixed-duration treatment of 1 year [13].

Indeed, Jansen et al. [14] showed that among patients who had a CR on anti-PD1, disease progression was significantly higher for those who received treatment duration of less than 6 months. Interestingly, further treatment beyond 6 months may not provide additional clinical benefit. Notably, this peculiar pharmacokinetics or optimal time frame for a favorable therapeutic ratio with anti-PD1 is more aligned with conventional chemotherapy than with genuine immunotherapy.

Again, we propose that clinical observations and clinical practice (so far) implicate that much of anti-PD1/L1’s therapeutic benefits may be achieved by means of anti-cancer rather than through immune modulatory effects. 

## 4. A Tale of Two Checkpoint Inhibitors

### 4.1. PD1/L1

Although we are inclined to investigate abnormal PDL1 expression in cancer cells and more specifically in CSC, perhaps its expression in normal SC is just as important and informative, especially if CSC mimic or mirror normal SC, and when the former may be derived from or is related to the latter. 

For example, PDL1 also plays an essential role in the maintenance of non-immunological functions such as epithelial-to-mesenchymal transition (EMT), glucose and lipid metabolism, stemness, and autophagy [15] of normal hematopoietic stem cells [16], mesenchymal stem cells (MSC) [17,18], mammary stem cells [19], skin stem cells [20] and induced pluripotent stem cells (iPSCs) [21]. 

Furthermore, the PD1/L1 axis provides immune evasive/protective capabilities for normal SC. After all, we need those very cells for the propagation, prolongation, and preservation of germline and reserve cells for both functioning and surviving. This is evident in the role of PDL1 for the prevention of immune rejection of embryonic-stem-cell-derived allograft [22] and iPSC-derived islet-like organoid [21]. 

However, if CSC is derived from or related to normal SC, then the same mechanisms of immune privilege and protection may be applicable to both cell types. It also implies that anti-PD1/L1 is more than immunotherapy; it targets certain tumor subtypes with specific phenotypes, such as EMT with embryonic mechanism and metabolism, stem-like processes, and stemness pathways. Paradoxically, anti-PD1/L1 may be precise because it targets stemness in a whole cell, but imprecise when it targets only the immune aspect of PD1/L1 in one part of the cell. Perhaps that is the reason we often need to combine treatments in multimodal therapy (rather than targeted therapy) and practice integrated medicine (rather than precision medicine) to maximize clinical outcome and optimize patient care.

So, how do we design multimodal therapy and integrated medicine in drug vs. therapy development? Again, the right cancer theory makes all the difference with respect to the direction and conduct of current cancer research and future cancer care. If we believe in immune activation/modulation, we combine anti-PD1/L1 with other immunotherapies. However, if we believe in anti-cancer therapy, we combine anti-PD1/L1 with alternative, complementary, or supplemental anti-cancer treatments. 

### 4.2. LAG3

As an immune CPI, LAG3 inhibits the activation of its host immune cell and generally promotes a more suppressive immune response. However, because it partners with other ligands (besides MHC II), such as liver and lymph node sinusoidal endothelial cell C-type lectin (LSECtin/CLEC4G), Galectin3 (Gal-3), Fibrinogen-like protein 1 (FGL1), all of which engender EMT and stemness phenotypes, we propose that its therapeutic efficacy could, in part, be due to the results of its direct anti-CSC effects versus indirect immune modulatory effects against CSC. 

Specifically, LSECtin, a transmembrane protein highly expressed on tumor-associated macrophages (TAMs), enhances stemness of breast cancer cells [23]. Loss of Gal-3 in at least some breast cancers, appeared to be associated with EMT and cancer stemness-associated traits, and predicted poor response to chemotherapy and poor prognosis [24]. Similarly, loss of FGL1 induced EMT in lung cancer [25], and expression of FGL1 in circulating tumor cells (CTCs) indicated poor prognosis in hepatocellular carcinoma (HCC) [26]. 

In the RELATIVITY-047 trial, those patients with advanced, previously untreated melanoma who were randomized to receive relatlimab plus nivolumab versus nivolumab alone experienced improved PFS—10.2 vs. 4.6 months [27]. The results suggest that targeting LAG3 may be clinically beneficial. But are the therapeutic effects of targeting LAG3 through direct cancer killing or by indirect immune modulating? That is the question.

## 5. Anti-Cancer and/or Immune Modulation

In many respects, whether anti-PD1/L1 is superior with to without anti-CTLA4 is key to the hypothesis that current immunotherapy is efficacious due to its effects on immune modulation and can be further improved with supplementary and complementary immune activating/modulatory strategies. 

A contrary hypothesis is that efficacy of anti-PD1/L1 and anti-LAG3 may, in fact, be due to their anti-cancer rather than immune modulatory effects, in which case anti-PD1/L1 plus another anti-cancer therapy would be preferable over anti-PD1/L1 plus another immune activating/modulatory therapy.

Ahn et al. [28] showed that a newly developed anti-PD-L1 antibody could induce antibody-dependent cellular cytotoxicity in myeloma cells. Collectively, they have developed a new anti-PD-L1 antibody that binds to mouse and human PD-L1 and demonstrated its antitumor effects in several syngeneic murine myeloma models. Apparently, the mode of action for this novel anti-PD-L1 antibody differs from that of existing anti-PD-L1 antibodies.

Or perhaps not. It is conceivable that some anti-PD1/L1 are directly tumoricidal or indirectly cytotoxic to CSC. After all, PDL1 is a stemness/EMT regulator [15] in several normal adult stem cells [16,17,18,19,20,21]. In contrast, anti-CTLA4 is mostly and purely immune modulating like a vaccine, IL-2, and IFN-alpha. Perhaps anti-PD1/L1 as an immune modulator is conventional thinking and the prevalent viewpoint. Whether it is a direct or indirect anti-cancer drug is an alternative perspective and narrative with immense implications for cancer research and patient care. 

Importantly, whether anti-PD1/L1 owes its therapeutic benefits to direct or indirect cytotoxic effects (as Ahn et al. showed) or immune modulatory effects (as most scientists and clinicians accept and assume) has therapeutic implications when it concerns combination strategy in multimodal therapy and integrated medicine for cancer care. Would it be more effective and safer by combination with another anti-cancer agent vs. another immune activator or modulator? 

## 6. Immune Activation/Modulation

### 6.1. Animal Models

It is of interest that a complete knockout of LAG-3 in mice displayed normal immune function [29]. Similarly, PD-1 and B7-H1/PD-L1 KO mice did not develop spontaneous autoimmune responses in the first year of age. However, they did develop late-onset, strain-specific autoimmunity: on a C57/Bl6 background, it manifested as sporadic glomerulonephritis [30], on a Balb/c background, as an antibody-mediated cardiomyopathy [31].

This is in stark contrast to those CTLA-4 knockout mice in which massive polyclonal expansion of T cells and multiorgan tissue destruction caused death at age 2 to 3 weeks [32]. Similarly, mice deficient for another purely immune modulating molecule, namely, IL-2, became severely immunocompromised and died between 4 and 9 weeks after birth [33]. Furthermore, 100% of those remaining surviving mice developed an inflammatory bowel disease with striking clinical and histological similarity to the human version of ulcerative colitis [34].

### 6.2. Pregnancy

Pregnancy is an obvious but often omitted experiment of nature when we consider how the immune system and how various immunotherapies affect the fetus. In many respects, the fetus is a “benign” tumor loaded with stem-like cells and tagged with abundant neoantigens, which are supposed to be more non-self than self to a mother’s otherwise intact immune system (compared to her own “malignant” tumor, should she develop one).

Hence, vaccines and allergy shots that utilize exogenous antigens to activate the immune system are generally safe during pregnancy. An exception is live attenuated vaccines like MMR and varicella, which can cross the placenta and may theoretically cause harm to the fetus, if not to the mother, when the vaccine or the immune system is somehow or becomes amiss. 

Importantly, use of anti-PD1 (e.g., nivolumab) is discouraged during pregnancy (category D), because it is associated with increased incidence of pregnancy complications [35,36,37]. Interestingly, anti-CTLA4 (e.g., ipilimumab), IFN alfa-2a, and IL-2 may be slightly safer than anti-PD1 for the fetus during pregnancy (category C). 

What is the reason for a subtle difference (if there is one) in the safety of anti-PD1 versus anti-CTLA4, interferon alfa-2a, or IL-2 to the fetus during pregnancy? In the case of IFN alpha, perhaps it is safer because it does not cross the placenta and does not inhibit DNA synthesis. In fact, oncologists would rather use IFN alpha than a tyrosine kinase inhibitor (TKI), such as imatinib (which is also category D), for pregnant patients who are symptomatic and who need treatment for their chronic myelogenous leukemia [38]. 

We surmise that when anti-cancer treatments, such as anti-PD1/L1 and TKI, target cancer and CSC but may also damage the embryo and normal SC (e.g., MSC express PDL1, morphogenetic pathways involve tyrosine kinases), they may pose more risk to the fetus than pure immune modulatory agents (such as anti-CTLA4, IFN-alpha, IL-2) do, because the latter do not cause harm to normal SC and are less likely or unlikely to inflict injury to the developing fetus.

### 6.3. Clinical Trials

It is evident that CPIs are not equal when it concerns clinical efficacy in cancer care. Whether a CPI is more clinically efficacious because it also exerts anti-cancer effects rather than just elicits immune modulatory effects needs further investigation. For example, unlike anti-PD1/L1 and anti-LAG3, anti-IDO1 and anti-TIGIT, just like IL-2 agonist, have been less stellar in randomized clinical trials so far (Table 1 and Table 2).

#### 6.3.1. IDO1

The phase-III ECHO-301/KEYNOTE-252 trial randomized patients with metastatic melanoma to receive either an IDO1 selective inhibitor (epacadostat) plus pembrolizumab or pembrolizumab plus placebo [40]. Unfortunately, the experimental combination did not improve outcomes (PFS: 4.7 vs. 4.9 months), compared to the control arm. 

#### 6.3.2. TIGIT

TIGIT is expressed in tumor-antigen-targeting T cells. An anti-TIGIT antibody (tiragolumab) failed to improve PFS when combined with the anti-PD-L1 antibody, atezolizumab, with (for SCLC, SKYSCRAPER-02) [44] or without (for NSCLC, SKYSCRAPER-01) chemotherapy.

#### 6.3.3. IL-2

The phase 3 PIVOT-09 study unexpectedly failed to improve the likelihood of response or OS with nivolumab combined with an engineered IL-2 pathway agonist (bempegaldesleukin), compared with a control TKI in patients with untreated renal cell carcinoma (RCC) [61]. This is noteworthy because PIVOT-09 was widely expected to be a positive study based on prior phase 1/2 results.

A separate study (PIVOT-10, phase 2) of combination treatment also failed to show benefits in cisplatin-ineligible, locally advanced, or metastatic urothelial bladder cancer, including those with low PD-L1 expression. Consequently, clinical development of bempegaldesleukin/nivolumab was terminated.

## 7. Checkpoint Inhibitors Are Not Equal

### 7.1. CTLA-4

One way to determine whether anti-CTLA4 is efficacious due to its immune modulatory effects is to examine whether it provides any tangible or meaningful clinical benefits (1) beyond the presumed anti-cancer (+/− immune modulatory) effects of anti-PD1/L1 (i.e., whether anti-CTLA4 + anti-PD1 is superior to anti-PD1 alone) and (2) compared with other purported anti-cancer treatments (e.g., whether anti-CTLA4 + anti-PD1 is superior to anti-PD1/L1 + chemotherapy for non-small-cell lung cancer/small cell lung cancer (NSCLC/SCLC), or anti-PD1/L1 + TKI for RCC) [76].

For the purposes of this discussion, we focus on published randomized clinical trials in selected malignancies (Table 1 and Table 2) and propose that comparing anti-CTLA4 + anti-PD1 with standard of care rather than out-of-date regimens in the control arm would be informative, e.g., with anti-PD1/L1 + chemotherapy rather than chemotherapy alone for NSCLC/SCLC, and with anti-PD1 + axitinib, lenvatinib, or cabozantinib rather than sunitinib alone for RCC, and could be practice-changing, if not paradigm-shifting. 

### 7.2. Melanoma-Advanced

In their meta-analysis, Serritella and Shenoy [77] found that ipilimumab + nivolumab compared with nivolumab alone did not provide tangible or meaningful improvement in OS and PFS for patients with NSCLC (squamous), NSCLC (PDL1 ≥ 1%), SCLC, pleural mesothelioma, urothelial carcinoma, esophageal carcinoma, sarcoma, or glioblastoma multiforme. 

The exception is melanoma. If this is so, why is melanoma an exception? Perhaps melanoma is a model malignancy for immunotherapy: it is exquisitely sensitive to immunotherapy, because it tends to have high tumor mutation burden (TMB) and more tumor neoantigens [78]. 

Although anti-CTLA4 + anti-PD1 clearly provides superior OS benefit in advanced melanoma compared with anti-PD1 alone [39,79], it is still unclear whether a pure immune modulatory drug such as anti-CTLA4 is better than a supposedly anti-cancer drug, such as BRAF/MEK inhibitor, when combined with anti-PD(L)1 for the treatment of advanced melanoma [80].

We propose that one way to ensure a favorable therapeutic ratio with substantive superior clinical outcome is to elevate drug vs. therapy development based on improved patient selection of appropriate tumor subtypes/phenotypes, according to a proper cancer theory and our scientific understanding of whether anti-cancer versus immune modulatory effects are key to optimal cancer care. 

### 7.3. Melanoma-Adjuvant

According to Atkins [81], the combination approaches that have been successfully developed to date have mostly included agents that have clinical activity as monotherapies. However, the clinical value of single agent anti-CTLA4 seems rather modest at best. Hence, the anti-CTLA4 antibody ipilimumab (10 mg/kg) plus dacarbazine significantly prolonged overall survival (11.2 vs. 9.1 months; HR 0.72; *p* < 0.001) compared to dacarbazine alone for patients with untreated metastatic melanoma [82]. But another anti-CTLA4 antibody, tremelimumab, failed to demonstrate survival benefit when compared to dacarbazine or temozolomide (12.6 vs. 10.7 months, HR, 0.88; *p* = 0.127) [83]. 

Therefore, whether anti-CTLA4 is the best option to combine with anti-PD1/L1 for the treatment of melanoma and other malignancies is a critical question that remains to be answered. It highlights a foundational question in drug vs. therapy development in cancer care (for the treatment of melanoma and other malignancies) whether the optimal strategies to combined anti-PD(L)1 treatments should be directed to or guided by anti-cancer vs. immune modulatory modalities.

For example, the IMMUNED trial was the first double-blind, randomized, placebo-controlled phase II trial to study the impact of a combination of ipilimumab and nivolumab compared to nivolumab monotherapy or placebo in patients with resected stage IV melanoma. Results demonstrated a significant benefit in recurrence-free survival (RFS) of combination ipilimumab and nivolumab compared to placebo with a 1-year RFS of 75% and 32%, respectively. However, the combination therapy of ipilimumab and nivolumab did not show a statistically significant benefit compared to nivolumab monotherapy, with 1-year RFS of 75% (CI 61.0–84.9%) and 52% (CI 38.1–63.9%), respectively [84].

Updated results of the IMMUNED trial confirmed that the addition of ipilimumab to nivolumab for adjuvant treatment did not demonstrate improved RFS compared to nivolumab monotherapy (2-year RFS of 64.6% and 63.2%, respectively) [85].

### 7.4. SCLC-ED

It is plausible that Serritella’s meta-analysis will be bolstered by the inclusion of another anti-CTLA4, namely, tremelimumab [86]. For example, adding tremelimumab to anti-PDL1 did not improve the clinical outcome of patients with aggressive cancers such as small cell lung carcinoma, extensive disease (SCLC-ED), in which chemotherapy is still a mainstay treatment to controlling disease and palliating symptoms [43].

### 7.5. NSCLC-Metastatic

Again, Serritella’s meta-analysis is consistent with the results of Keynote-598, in which patients with NSCLC and TPS ≥ 50% did not experience benefit from pembrolizumab + ipilimumab compared to pembrolizumab + placebo [50].

Similarly, MYSTIC demonstrated that in NSCLC w/PDL1 ≥ 25%, durvalumab + tremelimumab was not better than chemotherapy and may be inferior to durvalumab alone [49]. 

Although POSEIDON reported that durvalumab plus chemotherapy with a short course of tremelimumab demonstrated a statistically significant and clinically meaningful improvement in OS and PFS compared to chemotherapy, it remains unclear whether tremelimumab + durvalumab (+chemotherapy) is superior to durvalumab (+chemotherapy), and if so, by how much is clinically meaningfully better [47].

Therefore, the answers we obtain often depend on the questions we pose. Similarly, the experiments we will perform depend on the hypothesis we plan to test. 

Hence, how we frame our cancer theory affects how we design cancer trials. If immune modulation is pivotal to cancer therapy in metastatic NSCLC, then we design anti-CTLA4 + anti-PD1/L1 compared with standard chemotherapy. On the other hand, if anti-cancer therapy is central to cancer therapy in metastatic NSCLC, then we contend that a different design comparing anti-CTLA4 + anti-PD1/L1 with anti-PD1/L1 alone or with anti-PD1/L1 + standard chemotherapy would be more defining and revealing [87].

### 7.6. NSCLC-Locally Advanced

In many respects, neoadjuvant trials provide us a unique opportunity to assess the response to and benefit from treatments, because we have tumor tissues to examine after a particular treatment. Specifically, pathological complete response (pCR) may be informative and can be used as a surrogate endpoint, especially if it predicts OS improvements.

For example, pCR with chemotherapy alone for resectable lung cancer is low (about 2%). Addition of nivolumab to chemotherapy increases pCR to 24% [88]. But addition of ipilimumab to nivolumab + chemotherapy does not appear to improve pCR (18.2% vs. 18.2%) [89].

After all, how we interpret the results of an experiment (clinical trial or basic research) depends on the design of the experiments to test a hypothesis, which may or may not be correct or relevant. Again, the answers we receive often depend on the questions we pose, and the results we obtain depend on the hypotheses we formulate, no matter what tools (e.g., statistics) we use [87,90]. 

### 7.7. Additional Malignancies

Importantly, combined durvalumab + tremelimumab has generally not been shown to be superior to durvalumab monotherapy [91]. Although preliminary results suggest that durvalumab combined with tremelimumab may be superior to traditional chemotherapies in some malignancies, such as head and neck squamous cell carcinoma (HNSCC), pancreatic, and biliary tract cancers, they are still immature for other malignancies, such as ovarian cancer, colorectal cancer, and sarcomas [69,70,71,72,73,74]. 

## 8. Tumor Subtypes and Phenotypes

Perhaps tumor heterogeneity will provide us with the clues to solve the origin and nature of cancer immunity and the challenges of cancer immunotherapy. We have attempted to explain disparate responses of different cancer subtypes to “immune hot” and “immune cold” phenotypes, and to the presence or absence of myriad unique immune cells. However, when we lack a unified theory of cancer, we may not be able to distinguish whether these observations merely demonstrate effects rather than causes, associations rather than causations, passengers rather than drivers, and markers rather than makers of cancer and of carcinogenesis. 

Sometimes, we learn from negative results. Although anti-PD1/L1 is likely to be effective in tumors with high PDL1 expression, why is it also effective in certain kidney and bladder tumor subtypes with low PDL1 expression? Importantly, why is it relatively ineffective in some malignancies, such as multiple myeloma and prostate cancer regardless of PDL1 expression?

Suppose PDL1 (like LAG3) is not only a CPI but also a stemness marker (unlike other CPI), could that explain why anti-PD1/L1 (and anti-LAG3) is a better anti-cancer treatment compared with those other CPI (e.g., anti-CTLA4, anti-IDO1, anti-TIGIT), which are bona fide immunotherapy? If PDL1 is a CSC marker, with stem-like properties and EMT signatures [15,16,17,18,19,20], then anti-PD1/L1 should be more effective for the treatment of certain cancer subtypes with those very CSC phenotypes and EMT pathways than without. It would be more effective for the treatment of progenitor cancer stem cells than progeny differentiated cancer cells in the same solid tumor with inherent intra-tumoral heterogeneity by targeting the right tumor compartments, components, and the microenvironment.

### 8.1. Melanoma

It is notable that the four genomic subtypes of melanoma (BRAF, NRAS, NF1, and triple wild type) do not have distinguishing histopathological features or sites of origin (cutaneous, acral, mucosal, and uveal) [92]. Furthermore, they all harbor somatic aberrations, i.e., *BRAF*, *NRAS*, and *NF1* driver alterations that activate mitogen-activated protein kinase (MAPK) on the same signaling pathway. 

Assuming that classifications of primary melanoma by gene expression are biologically relevant and clinically useful, why do they all seem to converge on the function of microphthalmia-associated transcription factor (MITF) [93,94,95], which is closely linked to plasticity or stemness of melanoma cells? Intriguingly, an important regulator of MITF activity, namely Wnt/beta-catenin, is widely expressed in a variety of cancers.

Nevertheless, it is evident that melanomas from non-chronically sun damaged (non-CSD) sites have a different genomic landscape compared with those from CSD tissues. It is also true that varying exposure to UV-induced mutagenesis may cause different genomic defects in the same subtype of melanoma. What remains unclear is whether distinct subtypes of melanoma arise from different lineages of melanocyte progenitor cells and depend on a unique microenvironment or different mutations may affect different lineages of melanocyte progenitor cells and their progression to malignancy and response to treatments [74].

Perhaps KIT mutation exposes this dilemma in our understanding of the origin and nature of melanoma when it concerns the melanoma subtypes. Carvajal [96] reported KIT mutation rates of 21%, 18%, and 16% in acral, mucosal, and cutaneous melanomas (American patients), whereas Kong [97] revealed 12%, 10%, and 12% (Chinese patients), respectively. A phase II study found that the ORR for patients with melanoma harboring mutated KIT and treated with a specific treatment targeting this mutation (e.g., imatinib) was 23% [98].

Notably, the prognosis of metastatic melanoma was universally poor prior to 2010 regardless of subtype: median survival ranged from 10 to 13 months for cutaneous, acral, and uveal melanomas and 9 months for mucosal melanoma [99,100]. Importantly, biological differences between melanoma subtypes have become clinically significant with recent advances in systemic treatments. Hence, the survival of patients with metastatic cutaneous melanoma treated with combined nivolumab + ipilimumab or nivolumab has increased to 72.1 and 36.9 months, respectively [39]. Similarly, the survival of patients with metastatic acral melanoma has improved to 31.7 months with anti-PD1 [101]. However, the median survival of patients with metastatic mucosal (12.4 months) and uveal (7.6 months) melanomas treated with anti-PD1 remain dismal [101,102]. 

So, what makes uveal melanoma but not cutaneous melanoma deadly in the post-CPI era is the question. Does the presence of mutations in the GNAQ or GNA11 gene, absence of mutations in the BRAF V600 gene, or association of mutation in the BAP1 gene with metastasis that belies its unique stemness origins in uveal melanoma bestow its distinct prognostic and/or predictive dispositions? 

### 8.2. Renal Cell Carcinoma (RCC)

Verbiest et al. [103] demonstrated that 20% of RCC belonging to the “inflamed” subtype ccrcc4 tends to display sarcomatoid differentiation and BAP1 mutation. Interestingly, sarcomatoid differentiation indicates an aggressive stem-cell-like phenotype and displays the characteristic EMT features. Induction of EMT upregulates PD-L1 expression in sarcomatoid RCC (sRCC) [104], as well as in claudin-low breast cancer [105] and in NSCLC [106].

For example, sRCC expresses higher levels of PD-L1 and PD1 compared with clear cell RCC (ccRCC) (54% vs. 17% and 96% vs. 62%, respectively). Coexpression of both PDL1 on tumor cells and PD1+ TIL was observed in 50% of sRCC vs. 3% of ccRCC [107,108].

It is of interest that sRCC did not show any difference in the mutational load amongst cancer-related genes within its epithelioid and sarcomatoid components [109]. In contrast, sRCC had a completely different molecular pathogenesis and distinctive mutational and transcriptional profiles compared to ccRCC [110]. These findings support the hypothesis that both epithelial and sarcomatoid components of sRCC originate from the same progenitor cell, but clonal divergence occurs during tumor progression. 

Importantly, these results suggest that blockade of the PD(L)1 axis could be particularly effective for the treatment of EMT-derived tumors, such as sRCC. Furthermore, clinical trials that recruit sufficient patients with such tumor phenotypes are more likely to reveal clinical benefits using anti-PD1/L1 therapy. We predict that anti-PD1/L1 may be more beneficial for the treatment of EMT-derived tumors, such as sRCC rather than ccRCC, according to the stem-cell theory of cancer. 

### 8.3. Bladder Cancer

Similarly, the TCGA studies demonstrated that 35% of muscle invasive bladder cancer (MIBC) in the “basal-squamous” subgroup tended to display squamous differentiation and express high PDL1, whereas 19% of MIBC that belonged to the “luminal infiltrated” subgroup expressed moderate PDL1 and EMT markers (TWIST, ZED1). Luminal infiltrated MIBC tended to respond to anti-PD1/L1, but not to neoadjuvant chemotherapy, while basal-squamous MIBC were likely to respond to both treatments [111].

Again, treating all MIBC with anti-PD1/L1 may not show clinical benefit unless the clinical trials enroll enough patients with “luminal-infiltrated” and “basal-squamous” phenotypes, who are more likely to benefit from such treatment. Unfortunately, statistical power may still manage to overcome our ignorance and show evidence of clinical benefit by the force of numbers so that we will have license to treat everyone including those who are less likely or unlikely to benefit from the treatment.

### 8.4. Multiple Myeloma

Multiple myeloma (MM) is rife with genetic defects and neoantigens. However, it comprises predominantly differentiated malignant plasma cells, which are less likely to express EMT features. Pembrolizumab did not improve PFS of patients with relapsed or refractory MM (KEYNOTE-183) or treatment-naive MM (KEYNOTE-185) [112,113]. Whether the negative results of pembrolizumab for the treatment of MM are due to imbalances in patient and disease characteristics or to the biology of this malignancy remains unknown. 

What is known is that high-risk MM subtypes (i.e., MF, PR, MS) [114] express NIMA (never in mitosis gene A)-related kinase 2 (NEK2), have low-PDL1 levels, and are less likely to respond to anti-PD1 therapy. Interestingly, NEK2 is a centrosome-mitotic kinase involved in asymmetric cell division [115] and displaying EMT features [116]. It is a stem-like marker with prognostic and predictive value that promotes aerobic glycolysis [117] and modulates drug resistance ALHD1A1 [118,119]. Therefore, association of NEK2 with stemness in MM is not related to PDL1 as in sRCC. Although the prognostic implications for NEK2 and PDL1 with stemness are similar, the therapeutic implications are not.

### 8.5. Prostate Cancer

Like MM, prostate cancer comprises mainly differentiated malignant cells which are less likely to express EMT features. Like malignant plasma cells that produce an elevated differentiated product, namely, monoclonal antibody, differentiated prostate cancer cells produce their own elevated differentiated biomarker, namely PSA. 

Understandably, treatment of prostate cancer patients with anti-PD1/L1 is less likely to be beneficial for a vast majority of patients who do not have the right tumor phenotype: about 3% have defective mismatch repair/microsatellite instability-high (dMMR/MSI-H) [120] and 6% have tumor mutation burden (TMB) > 10 [121]. Unsurprisingly, atezolizumab did not improve OS of patients with metastatic CRPCa [122].

However, specific molecular profiles (e.g., dMMR/MSI-H and TMB > 10) may be enriched in certain CRPCa subtypes [123], in which anti-PD1/L1 are more likely to provide clinical benefit by targeting the proper tumor compartment, component, and the microenvironment and eliciting either direct or indirect anti-cancer vs. immune modulatory effects on prostate CSC and its onco-niche. 

For example, Schweizer et al. [124] showed that 40% of patients with ductal prostate cancer (dPCa) had MMR gene alteration, of which 75% had evidence of hypermutation in their tumors. Subsequently, they confirmed that 49% of patients with dPCa had at least one DNA damage repair gene alteration, including 14% with MMR gene mutation and 31% with homologous repair mutation [125]. It remains unknown whether the existence of stemness or stem-like features (such as EMT) ingrained in a particular tumor subtype (such as dPCa) or certain CRPCa subtypes [123,126] renders them more susceptible to immunotherapy. 

## 9. Beyond Immune Modulation

When we observe an exceptional or extraordinary clinical response, investigating its mechanisms of action is of interest and likely to be useful and relevant. In contrast, when a treatment is not effective at all, exploring its mechanism of action is less interesting and likely to be moot rather than relevant. 

There is no doubt that anti-PD1/L1 is an effective and beneficial treatment against cancer. The question is whether anti-PD1/L1 is a genuine immunotherapy or an inadvertent anti-cancer therapy. The right answer to this key question influences how we design better treatments beyond anti-PD1/L1: how do we make a great treatment even better? Do we enhance immunotherapy by combining it with another immunotherapy or immune modulator, or do we improve anti-cancer treatment by combining it with a complementary anti-cancer therapy?

When we believe that immunotherapy could be a panacea in cancer care, we combine anti-PD1/L1 with another CPI (such as anti-CTLA-4, anti-IDO1, anti-TIGIT) and with agents that target immune factors (e.g., IL-2) and effectors (e.g., dendritic cells, T cells, macrophages, NK cells), etc. 

However, when we think that anti-PD1/L1 is an actual anti-cancer treatment, we may be more inclined to combine it with another anti-cancer therapy. Specifically, if anti-PD1/L1 is an anti-CSC therapy, it will be more effective when we combine it with a complementary anti-CSC therapy, such as an anti-TKI, anti-HER2, or anti-Nectin4, or with a supplementary anti-CSC therapy, such as an anti-metabolic treatment that modulates the CSC metabolism, and/or with an anti-inflammatory treatment that modulates the CSC niche, and just as importantly when we combine it with anti-non-CSC therapy, such as those that target differentiated cancer cells, e.g., AR-dependent, PSA-expressing prostate cancer cells. 

## 10. Drug vs. Therapy Development

Sometimes, drug development succeeds but for the wrong reasons. However, the success rate under such circumstances tends to be relatively low, and any clinical benefits likely to be minimal, marginal, and momentary; the treatments tend to provide incremental clinical benefit that is statistically significant but can be clinically underwhelming [127]. 

Hopefully, therapy development succeeds for the right reason, according to the right cancer theory. A right cancer theory makes meaningful impacts in cancer research and cancer care. It directs us to implement integrative medicine instead of precision medicine and employ multimodal therapy rather than targeted therapy to increase cure rate and prolong remission time, which is cost effective and risk aversive, thereby providing exponential rather than incremental clinical benefits and outcomes. 

According to the stem-cell theory of cancer, anti-cancer therapy targeting different aspects of cancer (e.g., different compartment, components, and the microenvironment) may be superior to immune modulatory therapy alone. Multimodal therapy that employs supplementary or complementary anti-cancer therapy that targets CSC and non-CSC may provide greater efficacy and safety than those that combine multiple immune activating/modulatory treatments.

Therefore, it would be more beneficial combining anti-PD1/L1 with another anti-cancer agent, such as chemotherapy for SCLC and TNBC, with TKI for RCC and HCC, and with anti-LAG3 or a BRAF/MEK inhibitor for melanoma. Combining anti-PD1/L1 with another immune modulatory agent, such as anti-CTLA4, anti-IDO1, anti-TIGIT, or an IL-2 agonist is likely to be less beneficial or not beneficial at all. It is also likely to cause more severe and serious side effects and complications, including long-lasting, if not permanent, autoimmune diseases, like what we have observed in the CTLA-4 and IL-2 knockout mice.

## 11. Combination Strategies

Because cancer tends to be complicated rather than simple (e.g., heterogeneous) with active networks of abundant targets and redundant pathways, combined therapy and multimodal strategy are likely to be more effective and beneficial in cancer care. But how we combine treatments to maximize efficacy and optimize safety depends on our knowledge and understanding about the origin and nature of cancer. From the clinicians’ standpoint, it is preferable to give the fewest drugs possible to those patients who need them and to get the best overall clinical outcome. From the drug companies’ stance, it may be advantageous to give as many drugs to as many patients as possible, so long as the statistics vouch for them and the guidelines certify them. 

### 11.1. Anti-PD1/PDL-1 Combined with Immunomodulatory Therapy


Melanoma


#### 11.1.1. Advanced

If combined immunotherapy has merits, melanoma is the ideal tumor to show them. As mentioned previously (Section 7.3), the combination approach that is likely to succeed has mostly utilized agents with clinical activity as monotherapies [81]. 

An exception to this rule is the case of high-dose IL-2 combined with ipilimumab, in which either treatment has proven individual immunotherapeutic activity but together scarcely provide any improved result, i.e., 1/9 (11%) patients with advanced melanoma experienced a partial response [128]. 

#### 11.1.2. Neoadjuvant

Perhaps the clinical stage does matter. Patel et al. [129] showed that in a neoadjuvant setting, pembrolizumab (anti-PD1) alone elicited a complete pathological response (pCR) in 21% of patients with resectable advanced melanoma. Would combining anti-PD(L)1 with another immunomodulator or another anti-cancer agent enhance therapeutic efficacy of neoadjuvant therapy for the management of resectable advanced melanoma? 

Indeed, nivolumab (another anti-PD1) + ipilimumab provided a high pathological response rate (pRR) as well as pCR rate in a neoadjuvant setting [130]. However, it is notable that less ipilimumab and more nivolumab (Arm B: 2× IPI 1 mg/kg + NIVO 3 mg/kg Q3W) provided a higher pCR of 57% than more ipilimumab (Arm A: 2× IPI 3 mg/kg + NIVO 1 mg/kg Q3W) with a pCR of 43% or less nivolumab (Arm C: 2× IPI 3 mg/kg Q3W followed immediately by 2× NIVO 3 mg/kg Q2W) with a pCR of 24% [130].

Interestingly, high-dose IFN alpha-2b has shown efficacy (prolongs 5-year RFS by 10%) in an adjuvant setting for high-risk melanoma (thick primary and LN+), although any clinical benefit is qualified and seems to be modest [131]. Furthermore, the mechanisms of action for its antitumor activity remain unknown. In other words, it is still unclear whether IFN alpha-2b acts as an immunomodulator or is a direct anti-cancer agent against melanoma.

Nevertheless, the clinical efficacy of combined high-dose IFN alpha-2b and pembrolizumab in a neoadjuvant setting for patients with resectable regionally advanced melanoma seems to be clear: ORR, 73% and pCr, 43% [132]. What remains unclear is whether any synergistic activity in the combined treatment is due to their mutual immunomodulatory or direct anti-cancer effects.

Similarly, a single arm phase II trial (NeoPeLe) using pembrolizumab + lenvatinib showed a comparable partial pathological response of 75% (15/20 patients) and pCR of 40% (8/20 patients) [133]. In this scenario, lenvatinib is less likely to be immunomodulatory. It is also less likely to be directly anti-cancer compared with anti-BRAF plus anti-MEK treatment for patients with BRAF-mutated tumors [80]. Therefore, it is plausible that in a combined anti-PD(L)1-containing regimen, less anti-cancer activity may still be more efficacious than no anti-cancer and/or more immunomodulatory activity.

Again, the obvious and ultimate question is whether combining anti-PD1/L1 with another anti-cancer or immune activating/modulatory drug depending on the cancer subtype will provide a superior regimen with a more favorable therapeutic ratio in cancer care. 

### 11.2. Anti-PD1/PDL-1 Combined with Anti-Cancer Therapy

Chemotherapy

For tumors that are relatively chemo-sensitive, combining anti-PD1/L1 with chemotherapy makes sense, because both treatments elicit anti-cancer effects by different mechanisms of action, and the latter may expose neoantigens that enhance immune activation/modulation by the former treatment. 

It is conceivable that anti-PD1/L1 combined with chemotherapy is superior to anti-PD1/L1 with anti-CTLA4 for the treatment of certain cancers. When anti-PD1/L1 combined with chemotherapy becomes the standard of care, it will be appropriate to compare it with novel immunotherapy combinations to establish any new standards of care, while at the same time enable the elucidation of its basic mechanism of action, whether it is through anti-cancer versus immune activating/modulatory effects.


Small cell lung cancer


#### 11.2.1. Extensive Disease

The phase III IMpower133 study demonstrated improvement in PFS and OS in patients treated with four cycles of carboplatin, etoposide, and atezolizumab, followed by maintenance atezolizumab, compared to carboplatin and etoposide with placebo [41,134]. Similarly, the CASPIAN trial showed that durvalumab combined with chemotherapy improved OS over chemotherapy alone [42].


Non-small cell lung cancer


#### 11.2.2. Unresectable, Stage III

The PACIFIC trial showed improvement in PFS and OS with consolidation/maintenance durvalumab ×12 months for patients with stage II/III unresectable NSCLC that does not harbor a driver mutation such as EGFR or ALK and whose disease has not progressed after completion of concurrent chemoradiation therapy [45]. 

#### 11.2.3. Metastatic

KEYNOTE-189 trial showed that addition of pembrolizumab to standard chemotherapy comprising pemetrexed and a platinum-based drug in first-line metastatic non-squamous NSCLC resulted in significantly longer OS and PFS [46]. Similarly, KEYNOTE-407 demonstrated that pembrolizumab combined with carboplatin and a taxane was effective for the treatment of metastatic squamous NSCLC [48].


Triple-negative breast cancer (TNBC)


#### 11.2.4. Neoadjuvant, Stage II/III

KEYNOTE-522 demonstrated that the addition of pembrolizumab to a backbone of carboplatin/paclitaxel and anthracycline/cyclophosphamide improves pCR and 3-year event-free survival (EFS) (85% vs. 77%) for patients with higher-risk TNBC (cT2 or cT1N+) [51]. Notably, pCR and EFS improved, and irrespective of PDL1, CPS and EFS improved in both patients who achieved pCR and those with residual disease after neoadjuvant chemoimmunotherapy.

#### 11.2.5. Metastatic/Unresectable

Chemoimmunotherapy (paclitaxel/nab-paclitaxel or gemcitabine/carboplatin with pembrolizumab) was deemed a superior frontline option for PD-L1-positive (CPS > 10) metastatic TNBC patients compared to chemotherapy alone (KEYNOTE-355) [52]. Similarly, nab-paclitaxel with atezolizumab was approved in metastatic PDL1+ TNBC [53].

Table 1 and Table 2 illustrate additional, selected tumor types [64,66,67,68] and the clinical settings in which anti-PD(L)1 + chemotherapy have proven benefits and become the standard of care. It is by no means exhaustive because the list is rapidly expanding and may be never-ending.

### 11.3. Anti-PD1/PDL-1 Combined with Anti-CSC Agents

People may not realize that some of our more heralded targeted therapies, such as anti-HER2 and anti-Nectin4, may be effective and beneficial for a compelling reason: they target certain vital CSC properties. As expected, combining one anti-CSC therapy, such as anti-PD(L)1, with another, such as anti-HER2 or Nectin4, has an increased chance of providing an improved therapeutic ratio in drug vs. therapy development for cancer care. 

#### 11.3.1. BRAF+


Melanoma, advanced


Although anti-CTLA4 + anti-PD1 provides clinical benefit in advanced melanoma compared with anti-PD1 alone [39,79], it is still unclear whether combining a pure immune activating/modulatory drug such as anti-CTLA4 is better than a supposedly anti-cancer drug such as BRAF/MEK inhibitor with anti-PD(L)1 for the treatment of advanced melanoma.

After all, *BRAF*-mutant tumors confer a poorer prognosis relative to *BRAF* wild-type melanoma. BRAF-mutated melanoma tends to belong to MITF-low subgroup [93]. MITF is a regulator of melanocyte development/survival and melanin synthesis [135]. It inhibits invasion, inflammation, and EMT. In uveal melanoma, MITF loss is associated with loss of BAP1 protein expression, which is another stemness biomarker [136] associated with poor prognosis [137]. 

Interestingly, Boutros et al. [80] suggested that BRAF/MEK inhibitors + anti-PD(L)1 had a higher probability of achieving better PFS and ORR compared with ipilimumab + nivolumab for the treatment of advanced melanoma. However, further follow-up is required to perform an indirect comparison of response duration and OS.

#### 11.3.2. HER2+

Korkaya and Wicha [138] demonstrated that HER2 plays a central role in the intrinsic regulatory pathways of breast CSC through activation of the Wnt/beta-catenin pathway. HER-2 interacts with IL-8R (CXCR1/2) in the regulation of breast CSC by the tumor microenvironment [139,140]. Not surprisingly, the advent of anti-HER2 therapy has dramatically improved the clinical outcome of patients with HER2+ breast cancer [141]. It is of interest whether adding another anti-CSC therapy, such as anti-PD(L)1, to anti-HER2 will further improve overall clinical outcome.


Breast Cancer, Metastatic


A randomized phase 2 trial (KATE2) did not show PFS advantage in previously treated metastatic HER2+ breast cancer who received atezolizumab + T-DM1 vs. T-DM1+ placebo: 8.2 months vs. 6.8 months (HR 0.82, *p* = 0.33) [54].


Breast Cancer, Neoadjuvant


Impassion050 did not detect pCR improvement (ITT population: 62.4% vs. 62.7%; PD-L1+ cohort: 64.2% vs. 72.5%) in patients with high-risk (defined as >2 cm in size, node+) HER2+ breast cancer who received neoadjuvant dose-dense anthracycline, taxane, trastuzumab, and pertuzumab with or without atezolizumab [55].


Gastric Cancer, Metastatic


Currently, the standard of care for HER2-positive gastric cancer is pembrolizumab, trastuzumab, plus chemotherapy (either FOLFOX or capecitabine and oxaliplatin (CAPOX), according to KEYNOTE-811. ORR was higher in the pembrolizumab group (74 vs. 52%); patients also had a higher CR of 11% with pembrolizumab vs. 3% with placebo [65].

Intriguingly, the JACOB trial did not show improvement in ORR (57% versus 49%) or CR (5.7% vs. 2.0%) with the addition of pertuzumab to chemotherapy (CAPOX) and trastuzumab [142]. 

#### 11.3.3. Nectin4+


Urothelial cancer, metastatic


Enfortumab vedotin (EV), a Nectin-4-directed antibody-drug conjugate is effective and has been approved for the treatment of locally advanced or metastatic bladder cancer. 

Although Nectin-4 is a TIGIT ligand [143] and is involved in CPI like PD1, it is also a stem cell marker that upregulates EMT and metastasis, induces WNT/β-Catenin signaling via Pi3k/Akt axis [144], and cooperatively regulates with p95-ErbB2, the Hippo-signaling-dependent *SOX2* gene expression [145].

EV combined with pembrolizumab provides an ORR of 73% and a CR of 16% in previously untreated patients with metastatic urothelial carcinoma [146].

It will be a matter of time before one explores combining anti-PD1/L1 with other anti-CSC treatments, such as TROP2-, LILRB2-directed agents, et cetera.

### 11.4. Anti-PD1/PDL-1 Combined with TKI (Including Anti-VEGFR)

Another example of a stemness factor or target is the ubiquitous receptor tyrosine kinases (RTK), which play a vital role in pivotal embryonic and malignant processes, including pluripotency/differentiation, self-renewal/cell fate, morphogenesis, migration/invasion, and the like. 

Tyrosine kinase inhibitors (TKI) are known to be teratogenic because they disrupt embryogenesis; however, they also interfere with oncogenesis and are therapeutic in cancer care [147].

#### 11.4.1. Renal Cell Carcinoma

Although ipilimumab + nivolumab is effective for the treatment of intermediate and high-risk RCC compared with sunitinib, whether it is more effective than and just as safe as single agent nivolumab is less clear. 

Importantly, if anti-cancer trumps immune activating/modulatory effects, then one would expect that anti-PD1/L1 combined with TKI [56,57,58,60,148,149,150] will be superior to anti-PD1/L1 combined with anti-CTLA4 [59,151] for the treatment of intermediate-risk RCCcc (if not for poor-risk RCCcc and sRCC) [62,152] (Table 2).

#### 11.4.2. Hepatocellular Carcinoma

Similarly, tremelimumab + durvalumab showed superiority over sorafenib; however, whether it is superior to durvalumab alone is less evident in the HIMALAYA trial [63].

Although it is inappropriate to compare clinical trials because of different study designs and potential selection biases, one cannot help but discern no apparent superiority of tremelimuab and durvalumab over atezolimumab + bevacizumab in the IMbrave 150 trial (compared with the same sorafenib) [153]. 

### 11.5. Anti-PD1/PDL-1 Combined with Anti-Metabolic Agents

Because metformin targets CSC metabolism (e.g., glycolysis) and stemness pathways (e.g., EMT), it is expected to provide anti-cancer effects in the appropriate clinical setting, i.e., as preventive and maintenance therapy.

However, metformin may also elevate memory formation in T cells, enhance TIL infiltration, and protect TIL from apoptosis and exhaustion [154,155].

Therefore, our scientific knowledge and understanding which are derived from the scientific method and dependent on the hypotheses we formulate and the experiments we perform to test those hypotheses will ultimately determine whether any clinical benefit from metformin is due to its immune activating/modulatory or anti-cancer effects.

A pertinent scientific hypothesis is predicated by valid clinical observations. For instance, Ciccarese et al. [156] found a statistically significant correlation between higher doses of metformin (>1000 mg daily) and longer PFS (*p* = 0.021), longer OS (*p* = 0.037), and higher ORR in diabetic cancer patients who received concurrent nivolumab and metformin. 

A retrospective descriptive analysis carried out in the randomized phase III OAK trial for the treatment of advanced or metastatic previously treated NSCLC revealed ORR improvement in patients receiving concomitant atezolizumab and metformin [157]. 

Another retrospective cohort study including patients diagnosed with metastatic melanoma and treated with anti-PD-1 only or anti-CTLA4/anti-PD-1 combination therapies, with or without metformin, suggested favorable treatment-related outcomes including OS and PFS in patients who have received metformin in combination with immune CPI [158].

Hopefully, a well-designed prospective study will finally enable us to answer this seminal question of whether combined anti-PD1/L1 and an anti-metabolic drug (such as metformin) that targets CSC metabolism (as in CTC-mediated metastasis [159]) elicits anti-cancer and/or immune modulatory effects in cancer care.

### 11.6. Anti-PD1/PDL-1 Combined with Anti-Inflammatory Agents

Perhaps any subtle difference in the anti-cancer versus immune modulatory effects of curcumin, an anti-inflammatory food product, versus an authentic anti-inflammatory drug, is a matter of semantics and different perspectives of cancer. Perhaps a correct cancer theory and a proper cancer narrative matter.

It is plausible that curcumin exerts anti-cancer effects by attenuating the stemness STAT3 and NFkB pathways or mediates immune activating/modulatory effects by mitigating the same inflammatory factors STAT3 and NFkB in the tumor immune microenvironment (TIME). 

According to Paul and Sa [160], curcumin inhibits expression of PD-L1 and reduces the expression of CTLA4 and the number of Treg cells, thereby providing anti-cancer effects, if not enhancing salutary immune activating/modulatory results. 

Hayakawa et al. [75] showed that curcumin augments induction of tumor antigen-specific T cells by restoring the T cell stimulatory activity of DCs targeting activated STAT3 in both cancer cells and immune cells. Lim et al. [161] showed that NF-kB p65 induces COP9 signalosome 5 (CSN5) which is required for TNF-alpha mediated PD-L1 stabilization in cancer cells. Curcumin inhibits CSN5 and diminishes PD-L1 expression in cancer cells.

Interestingly, another anti-inflammatory agent like curcumin that targets STAT3 and NFkB, namely celecoxib, may also provide anti-cancer effects, if not immune modulatory effects. In the PICC trial, patients with locally advanced colorectal cancer (dMMR or MSI-H) who received toripalimab (an anti-PD-1 mAb) with or without celecoxib had a pCR rate of 88% vs. 65% at surgery, respectively [162]. 

The above studies suggest that combining curcumin with PD-1/PD-L1 Ab is an attractive strategy in the development of effective anti-cancer vs. immunotherapy for the treatment of various cancers.

## 12. Conclusions

Feynman also professed, “If there is an exception to any rule, and if it can be proved by observation, that rule is wrong”.

Sometimes, we know that the rule must be wrong because there are just too many exceptions to the rule. We are confounded by non-conformity to common observations and regular realities. We are bewildered by contradictions to scientific predictions and expectations. 

Often enough, the data may be perfect, but the rule or the idea is perfidious. When data do not support a rule or an idea, are the data at fault, or does the rule or idea need change?

In this perspective, we query whether anti-PD(L)1 is a beneficial anti-cancer treatment beyond its immune modulatory capabilities. Although this is a simple question, the answer has profound implications for cancer research and in cancer care. Although it may trigger a minor change in the conventional wisdom of cancer immunotherapy, it may instigate a major challenge to our current conduct in drug vs. therapy development and standard practice in patient care. 

Importantly, we need to discern truth from myth regarding cancer immunity (Figure 1), so that we do not squander our resources, waste our energy and effort, and subject our patients to risky treatments that provide marginal benefits. Are we being dogmatic and quixotic rather than pragmatic and realistic regarding immunotherapy when we conduct cancer research and deliver cancer care?

According to stem cell origin and the nature of cancer, are we relying too much on the ignorance of cancer experts in our design and practice of cancer immunotherapy?

## Figures and Tables

**Figure 1 cancers-16-01151-f001:**
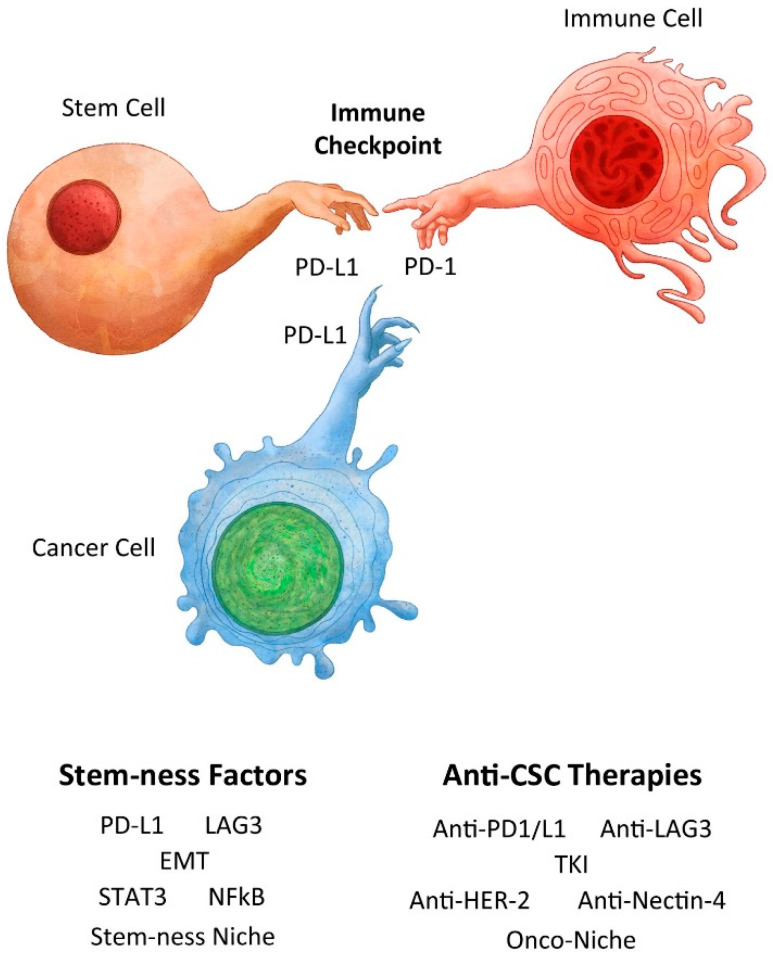
Creation of a normal stem cell and immune privilege with analogy to creation of a cancer stem cell and cancer immunity. Adapted from Michelangelo’s Creation of Adam by Benjamin Tu (www.bentubox.com) for this article (accessed on 14 January 2024).

**Table 1 cancers-16-01151-t001:** Selected randomized clinical trials combining anti-PD(L)1 with immune activating/modulatory agents or anti-cancer agents in melanoma, small cell lung cancer (SCLC), non-small-cell lung cancer (NSCLC), and breast cancer. A perspective on and review of published progression-free survival (PFS) and overall survival (OS) rather than strict comparison of studies, given differences in study designs and potential selection biases.

Cancer Type Stage Trial	Treatments	PFS or DFS (Months) or pCR (%)	OS (Months)	References (Randomized Phase II or III Trial)
Melanoma Advanced, metastatic untreated CHECKMATE 067	Nivo + ipi Nivo Ipi	11.5 6.9 2.9	72.1 36.9 19.9	Wolchok, 2022 [39] (III)
Melanoma Advanced untreated RELATIVITY 047	Nivo + relatlimab Nivo	10.1 4.6		Tawbi, 2022 [27] (II–III)
Melanoma, metastatic Anti-IDO1 ECHO-301/KEYNOTE 252	Pembro + epacadostat Pembro + placebo	4.7 4.9		Long, 2019 [40] (III)
SCLC-ED IMPower-133	Atezo + chemo Chemo	5.2 4.3	12.3 10.3	Horn, 2018 [41] (III)
SCLC-ED CASPIAN	Durva + chemo Chemo		13.0 10.3	Paz-Ares, 2019 [42] (III)
SCLC-ED CASPIAN	Durva + treme + chemo Durva + chemo Chemo		10.4 12.9 10.5	Goldman, 2021 [43] (III)
SCLC-ED untreated Anti-TIGIT SKYSCRAPER-02	Atezo + chemo + tiragolumab Atezo + chemo	5.4 5.6	13.1 13.1	Rudin, 2024 [44] (III)
NSCLC Unresectable, stage III PACIFIC	Durva after chemoRT ChemoRT	12.2 5.6	48 29	Antonia, 2018 [45] (III)
NSCLC Metastatic Non-squamous 1st-line KEYNOTE-189	Pembro + chemo Chemo	8.8 4.9	NR 11.3	Gandhi, 2018 [46] (III)
NSCLC Metastatic Non-squamous 1st-line POSEIDON	Durva + chemo + treme Durva + chemo	6.8 6.4	17.2 14.8	Johnson, 2023 [47] (III)
NSCLC Metastatic Squamous 1st-line KEYNOTE-407	Pembro + chemo Chemo	6.4 4.8	15.9 11.3	Paz-Ares, 2018 [48] (III)
NSCLC Metastatic Squamous 1st-line POSEIDON	Durva + chemo + treme Durva + chemo	4.6 4.7	10.4 11.5	Johnson, 2023 [47] (III)
NSCLC Advanced or metastatic 1st-line PDL1-high Anti-TIGIT SKYSCRAPER-01	Atezo+ tiragolumab Atezo		22.9 16.7	***
NSCLC Metastatic 1st-line ≥ PDL1 > 25% MYSTIC	Durva + treme Durva Chemo	3.9 4.7 5.4	11.9 16.3 12.9	Rizvi, 2020 [49] (III)
NSCLC Metastatic 1st-line PDL1 TPS > 50 KEYNOTE 598	Pembro + ipi Pembro	8.2 8.4	21.4 21.9	Boyer, 2021 [50] (III)
Breast cancer TNBC Neoadjuvant stage II/III KEYNOTE-522	Pembro + chemo Chemo	65% 51%		Schmid, 2022 [51] (III)
Breast cancer TNBC, CPS > 10 Metastatic or unresectable 1st line KEYNOTE-355	Pembro + chemo Chemo	9.7 5.6	23.0 16.1	Cortes, 2020 [52] (III)
Breast cancer TNBC PDL1+ Metastatic or unresectable 1st line IMpassion 130	Atezo + chemo Chemo	7.2 5.5	25.0 15.5	Schmid, 2018 [53] (III)
Breast cancer, HER2+, metastatic 2nd-line KATE2	Atezo + T-DM1 T-DM1	8.2 6.8		Emens 2021 [54] (II)
Breast cancer, HER2+, neoadjuvant Impassion050	Atezo + chemo + trastu + pertu chemo + trastu + pertu	62.4% (ITT) 64.2% (PDL1+) 62.7% (ITT) 72.5% (PDL1+)		Huober 2021 [55] (III)

DFS: disease-free survival; pCR: pathological complete response; SCLC-ED: small cell lung cancer-extensive disease; NSCLC: non-small-cell lung cancer; TNBC: triple negative breast cancer; nivo: nivolumab; ipi: ipilimumab; pembro: pembrolizumab, atezo: atezolizumab; durva: durvalumab; treme: tremelimumab; chemoRT: chemoradiation therapy; T-DM1: ado-trastuzumab emtansine; trastu: trastuzumab; pertu: pertuzumab; ITT: intention-to-treat. *** https://www.gene.com/media/press-releases/14998/2023-08-22/genentech-provides-update-on-phase-iii-s (accessed on 24 January 2024).

**Table 2 cancers-16-01151-t002:** Selected randomized clinical trials combining anti-PD(L)1 with immune activating/modulatory agents or anti-cancer agents in solid malignancies other than melanoma, SCLC, NSLC, and breast cancer. A perspective on and review of published PS and OS rather than strict comparison of studies, given differences in study designs and potential selection biases.

Cancer Type Stage Trial	Treatments	PFS or DFS (Months) or pCR (%)	OS (Months)	References (Randomized Phase II or III Trial)
RCC 1st line, advanced KEYNOTE-426	Pembro + axitinib Sunitinib	15.7 11.1	47.2 40.8	Plimack, 2023 [56] (III)
RCC 1st line, advanced CEAR	Pembro + Lenvatinib Sunitinib	23.3 9.2	NR, HR 0.72 NR	Choueiri, 2023 [57] (III)
RCC 1st line, advanced CheckMate 9ER	Nivo + cabozantinib Sunitinib	16.6 8.4	49.5 35.5	Burotto, 2023 [58] (III)
RCC 1st line, advanced CheckMate 214	Ipi + nivo Sunitinib	12.3, HR 0.86 12.3	55.7 38.4	Motzer, 2022 [59] (III)
RCC 1st line, advanced COSMIC-313	Ipi + nivo + cabozantinib Ipi + nivo	15.3 11.3		Choueiri, 2023 [60] (III)
RCC 1st line Advanced or metastatic Pegylated IL-2 PIVOT-09	Nivo + BEMPEG Tyrosine kinase inhibitor		29.0 NR	Tannir, 2022 [61] (III)
HCC HIMALAYA	Treme + durva Durva Sorafenib	3.8 3.7 4.1	16.4 16.6 13.8	Abou-Alfa, 2022 [62] (III)
HCC IMbrave 150	Atezo +bevacizum Sorafenib	6.8 4.3	19.2 13.4	Finn, 2020 [63] (III)
GEJ adenoca Adjuvant Checkmate 577	Nivo after preop chemoRT Preop chemoRT	22.4 11.0		Kelly, 2021 [64] (III)
Gastric or GEJ adenoca HER2+ metastatic KEYNOTE-811	Pembro + trastuzumab + chemo Trastuzumab + chemo	10.0 8.1	20.0 16.8	Janjigian, 2023 [65] (III)
Esophageal Squamous cell ca Metastatic KEYNOTE-590	Pembro + chemo Chemo	6.3 5.8	12.4 9.8	Sun, 2021 [66] (III)
Esophageal Squamous cell ca Metastatic PDL1 ≥ 1% Checkmate-648	Nivo + chemo Nivo + ipi Chemo	6.9 4.0 4.4	15.4 13.7 9.1	Doki, 2022 [67] (III)
Biliary tract cancer Metastatic TOPAZ-1	Durva + chemo Chemo	7.2 5.7	12.8 11.5	Oh, 2022 [68] (III)
Pancreatic cancer Metastatic	Durva + treme Durva	1.5 1.5	3.1 3.6	O’Reilly, 2019 [69] (II)
HNSCC Previously treated PDL1 ≥ 25% EAGLE	Durva + treme Durva SOC	2.0 2.1 3.7	4.8 9.8 9.0	Ferris, 2020 [70] (III)
HNSCC PDL1-low/neg CONDOR	Durva + treme Durva Treme		7.6 6.0 5.5	Siu, 2019 [71] (II)
HNSCC Recurrent or metastatic CHECKMATE 714	Nivo + ipi Nivo (platinum-refractory)	2.6 2.6	10.0 9.6	Harrington, 2023 [72] (II)
HNSCC Recurrent or metastatic CHECKMATE 714	Nivo + ipi Nivo (platinum-eligible)	2.8 2.9	10.0 12.9	Harrington, 2023 [72] (II)
Urothelial Untreated advanced or metastatic DANUBE	Dur + treme Durva Chemo		15.1 14.4 12.1	Powles, 2020 [73] (III)
Ovarian cancer Recurrent or persistent NRG	Nivo + ip Nivo	3.9 2.0	28.1, *p* = 0.43 21.8	Zamarin, 2020 [74] (II)
CRC Neoadjuvant Locally advanced	Toripa + celecoxib Toripa	88% 65%		Hu, 2022 [75] (II)

DFS: disease-free survival; pCR: pathological complete response; RCC: renal cell carcinoma; HCC: hepatocellular carcinoma; GEJ: gastro-esophageal junction; HNSCC: head & neck squamous cell carcinoma; CRC: colorectal carcinoma; nivo: nivolumab; ipi: ipilimumab; pembro: pembrolizumab, atezo: atezolizumab; durva: durvalumab; treme: tremelimumab; BEMPEG: bempegaldesleukin; toripa: toripalimab; chemoRT: chemoradiation therapy; trastu: trastuzumab; SOC: standard of care.

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
