# Peer review of "Stem Cell Origin of Cancer: Clinical Implications beyond Immunotherapy for Drug versus Therapy Development in Cancer Care"

_cancers, 2024, doi:10.3390/cancers16061151_

Round 1

Reviewer 1 Report

Comments and Suggestions for Authors

I read with great interest the article by Dr. Tu Shi-Ming and co-authors entitled «Stem Cell Origin of Cancer: Clinical Implications beyond Immunotherapy for Drug versus Therapy Development in Cancer Care». The manuscript is a continuation of previously published in Cancers articles and addresses the important topic of antitumor immunity and immunotherapy in the context of cancer stem cell theory postulating the origin of tumors from normal regional stem cells. The article is written as an authors’ opinion rather than a literature review, making it difficult to critically analyze.  In general, the article is written in good language and does not contain noticeable flaws. There are only 3 substantiated self-citations among 160 references. The drawback of the manuscript is that it is poorly illustrated. Adding colorful illustrations could attract the reader and make the text easier to understand.  The manuscript contains opinions and views with which I completely agree, and those that I do not share. However, since the article is intended to initiate discussion rather than inform readers, I believe it can be published without major modification. Just a few small notes should be taken into account.

1. Row 30 and below. The abbreviation “NSC” is commonly used to refer to neural stem cells. This may confuse the reader. It is better to change to “SC” or another one that is not used so often.

2. Table 1 and 2. 3rd column top line. There is a typo. “PFS or or DFS”.

3. All abbreviations for tumor names should be explained when they are first used and at the bottom of the tables. Also, it is not good to use acronyms as subheadings (row 271, 332, 633, 640, etc)

4. Row 652. TNBC is an acronym for Breast cancer. Probably “Triple negative breast cancer” is correct.    

Author Response

We thank the reviewers for your invaluable comments and suggestions. We have provided our responses below in bold with changes highlighted in red underneath the reviewer comments. We have incorporated all changes recommended by the reviewers with the changes highlighted in red in the manuscript.

Reviewer 1

I read with great interest the article by Dr. Tu Shi-Ming and co-authors entitled «Stem Cell Origin of Cancer: Clinical Implications beyond Immunotherapy for Drug versus Therapy Development in Cancer Care». The manuscript is a continuation of previously published in Cancers articles and addresses the important topic of antitumor immunity and immunotherapy in the context of cancer stem cell theory postulating the origin of tumors from normal regional stem cells. The article is written as an authors’ opinion rather than a literature review, making it difficult to critically analyze.  In general, the article is written in good language and does not contain noticeable flaws. There are only 3 substantiated self-citations among 160 references. The drawback of the manuscript is that it is poorly illustrated. Adding colorful illustrations could attract the reader and make the text easier to understand.  The manuscript contains opinions and views with which I completely agree, and those that I do not share. However, since the article is intended to initiate discussion rather than inform readers, I believe it can be published without major modification. Just a few small notes should be taken into account.

We thank the reviewer for your expert and insightful comments.

  1. Row 30 and below. The abbreviation “NSC” is commonly used to refer to neural stem cells. This may confuse the reader. It is better to change to “SC” or another one that is not used so often.

Agree! Changed NSC to normal SC, as recommended.

  1. Table 1 and 2. 3rd column top line. There is a typo. “PFS or or DFS”.

Typo corrected.

  1. All abbreviations for tumor names should be explained when they are first used and at the bottom of the tables. Also, it is not good to use acronyms as subheadings (row 271, 332, 633, 640, etc)

Done.

  1. Row 652. TNBC is an acronym for Breast cancer. Probably “Triple negative breast cancer” is correct.    

Yes.

Reviewer 2 Report

Comments and Suggestions for Authors

A high-valued perspective of the authors to reexamine whether effective immunotherapies are efficacious due to their anti-cancer and/or immune modulatory mechanisms. Several questions are listed as below,

1) According to “we propose that clinical observations and clinical practice (so far) implicate that much of anti-PD1/L1’s therapeutic benefits may be achieved by means of anti-cancer 1rather than through immune modulatory effects.”, the authors might provide more evidences or emphasize clearly in this paper. For example, anti-cancer is showed in 5.0. Anti-Cancer and/or Immune Modulation, in contrast, more details are described about immune modulation” in “6.0. Immune Activation/Modulation” section.

2) In  5.0. Anti-Cancer and/or Immune Modulation, it is claimed that “After all, PDL1 is a stem-ness/EMT biomarker [15]”, it should be noted that this reference only shows that PDL1 can regulating EMT signaling pathways, the word “biomarker” here might be not appropriate.

3) The survival of patients with advanced melanoma is still poor. One paper recently proposes a potential "Anti-Warburg Effect" (AWE) in CTCs-a metabolic shift that could be clinically important as they are significantly correlated with therapeutic response in melanoma patients (https://pubmed.ncbi.nlm.nih.gov/38300633/). Such advanced view should be better added.

4) What is “Phenotypes” specifically mentioned in “8.0. Tumor Subtypes and Phenotypes”?

Author Response

We thank the reviewers for your invaluable comments and suggestions. We have provided our responses below in bold with changes highlighted in red underneath the reviewer comments. We have incorporated all changes recommended by the reviewers with the changes highlighted in red in the manuscript.

Reviewer 2

A high-valued perspective of the authors to reexamine whether effective immunotherapies are efficacious due to their anti-cancer and/or immune modulatory mechanisms. Several questions are listed as below,

We agree with the reviewer that reexamination of whether effective immunotherapies are efficacious due to their anti-cancer and/or immune modulatory mechanisms is “high-valued” because it could be paradigm shifting and practice changing.

1) According to “we propose that clinical observations and clinical practice (so far) implicate that much of anti-PD1/L1’s therapeutic benefits may be achieved by means of anti-cancer 1rather than through immune modulatory effects.”, the authors might provide more evidences or emphasize clearly in this paper. For example, “anti-cancer” is showed in “5.0. Anti-Cancer and/or Immune Modulation”, in contrast, more details are described about “immune modulation” in “6.0. Immune Activation/Modulation” section.

One way to discern anti-cancer effect and/or immune modulation is to demonstrate differential therapeutic benefits among the CPI according to randomized clinical trials, as detailed in section 6.3.

Importantly, “whether anti-CTLA4 is the best option to combine with anti-PD1/L1 for the treatment of melanoma and other malignancies is a critical question that remains to be answered. It highlights a foundational question in drug vs therapy development in cancer care (for the treatment of melanoma and other malignancies) whether the optimal strategies to combined anti-PD(L)1 treatments should be directed to or guided by anti-cancer vs immune modulatory modalities.” (page 12, section 7.3, paragraph 2)

Serritella and Shenoy provided some of the most compelling evidence in this regard (section 7.2):

“Melanoma- advanced. In their meta-analysis, Serritella and Shenoy [43] found that ipilimumab + nivolumab compared with nivolumab alone did not provide tangible or meaningful improvement in OS and PFS for patients with NSCLC (squamous), NSCLC (PDL1 >1%), SCLC, pleural mesothelioma, urothelial carcinoma, esophageal carcinoma, sarcoma, or glioblastoma multiforme.

The exception is melanoma. If this is so, why is melanoma an exception? Perhaps melanoma is a model malignancy for immunotherapy…

Although anti-CTLA4 + anti-PD1 clearly provides superior OS benefit in advanced melanoma compared with anti-PD1 alone [45,46], it is still unclear whether a pure immune modulatory drug such as anti-CTLA4 is better than a supposedly anti-cancer drug such as BRAF/MEK inhibitor when combined with anti-PD(L)1 for the treatment of advanced melanoma [47].”

We have added another evidence and a recent publication by Santoni et al in this regard for patients with RCC (section 11.4.1, paragraph 2):

“…if anti-cancer trumps immune activating/modulatory effects, then one would expect that anti-PD1/L1 combined with TKI [140-146] will be superior to anti-PD1/L1 combined with anti-CTLA4 [147,148] for the treatment of intermediate-risk RCCcc (if not for poor-risk RCCcc and sRCC).”

Santoni M, Buti S, Myint ZW, Maruzzo M, Iacovelli R, Pichler M, Kopecky J, Kucharz J, Rizzo M, Galli L, et al. Real-world outcome of patients with advanced renal cell carcinoma and intermediate- or poor-risk international metastatic renal cell carcinoma database consortium criteria treated by immune-oncology combinations: Differential effectiveness by risk group? Eur Urol Oncol 2024: 7:102-11.

2) In “ 5.0. Anti-Cancer and/or Immune Modulation”, it is claimed that “After all, PDL1 is a stem-ness/EMT biomarker [15]”, it should be noted that this reference only shows that PDL1 can regulating EMT signaling pathways, the word “biomarker” here might be not appropriate.

Agree. Changed “biomarker” to “regulator”.

3) The survival of patients with advanced melanoma is still poor. One paper recently proposes a potential "Anti-Warburg Effect" (AWE) in CTCs-a metabolic shift that could be clinically important as they are significantly correlated with therapeutic response in melanoma patients (https://pubmed.ncbi.nlm.nih.gov/38300633/). Such advanced view should be better added.

Thank you for pointing out this important reference, which we have added to section 11.5 on anti-metabolic therapy, page 28, paragraph 6, lines 3-4.

Jiang Z, He J, Zhang B, Wang L, Long C, Zhao B, Yang Y, Du L, Luo W, Hu J, et al. A potential “anti-Warburg effect” in circulating tumor cell-mediated metastatic progression? Aging Dis 2024; doi:10.14336/AD.2023.1227.

4) What is “Phenotypes” specifically mentioned in “8.0. Tumor Subtypes and Phenotypes”?

Phenotypes refer to the observable trait of the subtypes biologically (e.g., EMT, heterogeneity) and clinically (e.g., prognostic and/or predictive dispositions).

Reviewer 3 Report

Comments and Suggestions for Authors

The current manuscript offers a compelling perspective on the potential mechanisms underlying the efficacy of immunotherapies, particularly their anti-cancer and/or immune-modulatory effects. These findings hold significant implications for the development of future combination therapy regimens for various types of cancer. The authors have effectively utilized previously published clinical trials to raise a critical question that warrants further investigation in future studies. 

Author Response

Reviewer 3

The current manuscript offers a compelling perspective on the potential mechanisms underlying the efficacy of immunotherapies, particularly their anti-cancer and/or immune-modulatory effects. These findings hold significant implications for the development of future combination therapy regimens for various types of cancer. The authors have effectively utilized previously published clinical trials to raise a critical question that warrants further investigation in future studies. 

We thank the reviewer for your inspiring and insightful comments.

Round 2

Reviewer 2 Report

Comments and Suggestions for Authors

No other questions.